# Plant-Based Diets and the Incidence of Asthma Symptoms among Elderly Women, and the Mediating Role of Body Mass Index

**DOI:** 10.3390/nu15010052

**Published:** 2022-12-22

**Authors:** Wassila Ait-hadad, Annabelle Bédard, Rosalie Delvert, Laurent Orsi, Sébastien Chanoine, Orianne Dumas, Nasser Laouali, Nicole Le Moual, Bénédicte Leynaert, Valérie Siroux, Marie-Christine Boutron-Ruault, Raphaëlle Varraso

**Affiliations:** 1Université Paris-Saclay, UVSQ, Université Paris-Sud, Inserm, Équipe d’Épidémiologie Respiratoire Intégrative, CESP, 94805 Villejuif, France; 2Team of Environmental Epidemiology Applied to the Development and Respiratory Health, Institute for Advanced Biosciences, Inserm U 1209, CNRS UMR 5309, Université Grenoble Alpes, 38000 Grenoble, France; 3Université Paris-Saclay, UVSQ, Université Paris-Sud, Inserm, Gustave Roussy, Équipe “Exposome et Hérédité”, CESP, 94805 Villejuif, France

**Keywords:** healthful plant-based diet index, unhealthful plant-based diet index, body mass index, obesity, incidence of asthma symptoms, asthma prevention, mediation analyses, counterfactual framework, smoking

## Abstract

We aimed to test the hypothesis that adherence to a healthful plant-based diet (hPDI) is associated with a subsequent decrease in the incidence of asthma symptoms, with an opposite association with adherence to an unhealthful plant-based diet (uPDI). In addition, we evaluated a potential mediating role of body mass index (BMI) and the modifying effect of smoking. Among 5700 elderly women from the French Asthma-E3N study with dietary data in 1993 and 2005, we assessed the incidence of asthma symptoms in 2018 among women with no asthma symptoms in 2011. BMI was evaluated in 2008. Mediation analyses in the counterfactual framework were used to disentangle total, direct, and indirect effects mediated by BMI. We found that both healthful and unhealthful plant-based diets were associated with a lower incidence of asthma symptoms over time, mediated by BMI (OR (95%CI) for the indirect effect: 0.94 (0.89–1.00) for hPDI and 0.92 (0.70–1.00) for uPDI)). Associations with both healthful and unhealthful PDIs were mediated by changes in BMI by 33% and 89%, respectively. Plant-based diets (healthful and unhealthful) were associated with subsequently reduced incidences of asthma symptoms over time, partly or almost totally mediated by BMI according to their nutritional quality.

## 1. Introduction

Asthma is one of the most common chronic diseases, affecting roughly 262 million people in the world [1]. It is a heterogeneous disease with different phenotypes and symptoms that may vary over time [2]. Among adults over 65 years, the worldwide prevalence of asthma ranges from 4–13% [3], a rate probably underestimated because of frequent underdiagnosis in that age group [4]. The burden of asthma is higher in the elderly than in younger adults [5,6], with higher rates of hospitalizations [4,6] and poorer health-related quality of life [3,4,6]. Therefore, asthma in the elderly is a phenotype of interest, especially in older women, where it tends to be more prevalent and more severe than in older men [7]. 

Investigating the role of modifiable lifestyle factors such as diet for the primary prevention of this highly prevalent disease is warranted, especially among older women. Indeed, epidemiological studies on the effect of diet on asthma outcomes have widely focused on children and young to middle-aged adults [8]. Overall, these studies suggested a beneficial impact of certain plant foods such as fruit and vegetables [9] or dietary fiber from whole grains [10] and a deleterious impact of animal foods such as processed meats [11]. However, studying foods only raises several conceptual and methodological concerns since we consume foods in the form of complex combinations (i.e., meals), which leads to interactions between foods and influences the bioavailability and absorption of nutrients. Therefore, considering dietary patterns and, more particularly, dietary scores, which enables accounting for correlations and interactions between foods and nutrients, is likely a more accurate approach to evaluating the impact of dietary habits on asthma [12].

The term “plant-based diets” encompasses a wide range of dietary patterns that contain low amounts of animal products and high amounts of plant products and are becoming increasingly popular [13,14]. It has been shown that a plant-based diet that favors “healthy” plant foods (such as whole grains, fruit, and vegetables) over less-healthy plant foods (such as refined grains), as evaluated by the “*healthful* Plant-based Diet Index” (hPDI), was associated with substantially lower risk of chronic diseases such as diabetes and coronary heart disease (CHD); in the opposite, a plant-based diet which favors less-healthy plant foods over healthier plant-foods, as estimated by the “*unhealthful* Plant-based Diet Index” (uPDI), has been found associated with higher rates of chronic diseases [15,16]. Therefore, distinguishing plant foods based on their nutritional quality seems essential, and despite the apparent protective effect of plant foods such as fruit and vegetables [9] against asthma risk, it remains unknown whether the overall intake of plant foods is beneficial. 

Obesity is now an established risk factor for asthma [17,18], with recent studies supporting the hypothesis that obesity is causally related to asthma [19], especially in older women [20]. As better diet quality has been associated with a lower risk of obesity [21], obesity could lie in the causal pathway between diet and asthma, and body mass index (BMI) could act as a mediator rather than a confounder in this association [22]. Therefore, longitudinal data are needed to properly disentangle the direct effect of diet on asthma incidence (i.e., independently of BMI) from its indirect effect (i.e., explained by BMI). To our knowledge, no study has investigated the association between plant-based diets and the incidence of asthma symptoms. A previous study in middle-aged adults reported that better diet quality, evaluated through the Alternate-Healthy Eating index-2010 (AHEI-2010), was associated with improved asthma symptoms over time, and the association was not mediated by BMI [23]. In addition, that study reported the association only in never smokers, suggesting a modifying effect of smoking. Indeed, diet may also interact with other factors, such as smoking, that could modify the pulmonary oxidant/antioxidant balance [24]. Therefore, addressing a potential modifying effect of smoking in the diet–asthma incidence association is further warranted. 

Our aims were to determine whether adherence to the healthful version of the plant-based diet (e.g., hPDI) would be associated with a decreased incidence of asthma symptoms, whereas adherence to the unhealthful version (e.g., uPDI) with an increased incidence of asthma symptoms, while accounting for a potential mediating role of BMI and a modifying effect of smoking.

## 2. Materials and Methods

### 2.1. Study Population

The E3N study (Etude Epidémiologique auprès des femmes de la Mutuelle Générale de l’Education Nationale [MGEN]) is a prospective cohort conducted in 1990 among 98,997 women enrolled in a French national health insurance plan covering mostly teachers [25]. Since 1990, information on lifestyle and medical history has been collected approximately every two years by means of self-administered questionnaires, and dietary information was collected twice by validated semi-quantitative food history questionnaires (1993 and 2005). 

Participants were women from the asthma-E3N study, a nested case-control study on asthma within the E3N cohort. Briefly, 7100 women who answered positively to the single question “Have you ever had an asthma attack?” in the main E3N questionnaires at least once between 1992 and 2008 were selected as “asthma cases,”; and 14,200 women who never reported any asthma attack between 1992 and 2008 as “age-matched women without asthma.” They have been contacted to fill in a validated questionnaire on respiratory health. [26,27] (Appendix A). Out of those 21,300 women, 19,404 completed the respiratory questionnaire (91% response rate), allowing us to evaluate the asthma symptoms score in 2011. In 2018, through the main E3N follow-up questionnaire, new data were collected regarding the asthma symptoms score, and out of these 19,404 women, 15,301 returned in 2018 (Q12). 

We applied several exclusion criteria: first, women who answered the asthma symptoms scored neither in 2011 nor in 2018 (n = 5004). Then, women who completed the dietary questionnaire neither in 1993 nor in 2005 or had an implausibly high (top 1% of the ratio between energy intake and energy requirement (EIER)) or low (bottom 1% of the EIER ratio) total energy intake in 1993 or in 2005 (n = 1676). Finally, and as our main aim was to investigate the incidence of asthma symptoms over time (and not prevalent asthma symptoms), women who reported at least one asthma symptom in 2011 were excluded (n = 2921). Our analytic population included 5700 women. A comparison of excluded vs. included participants is presented in Appendix A. 

The French Institutional Ethics Committee approved the study protocol, and all participants provided their written informed consent.

### 2.2. Diet Assessment

Two validated semi-quantitative food history questionnaires administered in 1993 and 2005 were used to collect dietary data. The first part of the questionnaire, a semi-quantitative questionnaire, assessed consumption frequencies and portion sizes for eight potential daily meals ((1) breakfast, (2) morning snack, (3) aperitif before lunch, (4) lunch, (5) afternoon snack, (6) pre-dinner aperitif, (7) dinner, and (8) after dinner snack) for 66 foods or food groups. Frequency was expressed in 11 potential categories: never or less than once a month; 1, 2, or 3 times a month; and 1 to 7 times a week. A photo booklet was also sent to facilitate the estimation of portion sizes [28]. The second part of the questionnaire was qualitative and allowed detailed consumption of specific foods within the food groups mentioned in the first part of the questionnaire. 

As previously described, we derived the plant-based diet index (PDI) and the healthful and unhealthful versions of the score [16] for each food history questionnaire (1993 and 2005) based on the intake of 18 food groups (Appendix A). As limited information was available for whole grains intake, we used data from fiber from cereal products instead. Food groups were ranked into quintiles of consumption, or a non-consumer category (if concerning over 20% of women) plus quartiles among consumers; for creating the PDI, positive scores were given to plant food groups, and reverse scores to animal food groups; for creating the hPDI, positive scores were given to healthful plant food groups and reverse scores to unhealthful plant food groups and animal food groups; for the uPDI, positive scores were given to unhealthful plant food groups and reverse scores to healthful plant food groups and animal food groups. The 18 food group scores were summed up to obtain the indices (ranging from 18 to 90), and high scores for the three indices reflected low animal food intakes. We used the mean of the 1993 and 2005 scores categorized either into quintiles when sample size allowed it or into tertiles. The correlation between hPDI and uPDI was 0.27.

### 2.3. Assessment of Asthma Symptoms Incidence 

The asthma symptoms score, a previously validated continuous measure of asthma in epidemiological studies, was used to evaluate the incidence of asthma symptoms over time [29,30]. Ranging from 0–5, the score is based on the number of respiratory symptoms during the past 12 months: breathless while wheezing; woken up with chest tightness; attacks of shortness of breath at rest; attacks of shortness of breath after exercise; woken up by attacks of shortness of breath. Women with no asthma symptoms in 2011 and in 2018 served as the reference group (n = 5149), and women with no symptoms in 2011 and at least one symptom in 2018 as “incident” (n = 551). 

### 2.4. Body Mass Index 

Height was collected in 1990. Current weight was collected at each questionnaire since inclusion. BMI was calculated from self-reported current weight and height in 2008, expressed in kg/m^2^, and analyzed as a continuous variable. BMI was also calculated in 2005 and 2011.

### 2.5. Other Variables

Socio-demographic and lifestyle characteristics were collected by self-administered questionnaires. Physical activity was assessed in 2005 using data from several questions on different activities [31] and expressed in metabolic equivalent of tasks (METs) per week (MET-hours/week). Smoking status assessed in 2005 was categorized into 3 classes: never, former, or current smoker. Educational level collected in 1992 was categorized into 4 classes: < high school diploma, high school to 2-level university, 3–4-level university, and ≥ 5-level university. Marital status (married: yes or no) was collected in 1990. Having farmer parents was collected in 2007 [32]. 

### 2.6. Statistical Analyses

The “last observation carried forward” method was used to impute missing data. We used a graph to represent our mediation model and hypotheses (Figure 1). 

Several methods of mediation analysis in a counterfactual approach have been proposed to disentangle the direct and indirect effects in the longitudinal context [33]. We applied marginal structural models, as proposed by Lange et al. [34], to dissociate the direct effect of a plant-based diet on the incidence of asthma symptoms (OR_DE_) from the indirect effect mediated by BMI (OR_IE_). The mediation analysis was implemented with the following steps: (i) for PDI scores considered as categorical, we created a new data set by repeating each observation from the original data set, five times for quintiles and three times for tertiles, and by including a new variable A*, equal to the original exposure for the first replication, and equal to all the possible values for the other replications; for PDI scores considered as continuous, we created a new data set by repeating each observation from the original data set 10 times (a minimum of 5 draws being recommended [34]), and by including a new variable *A**, equal to the original exposure for the first replication, and equal to randomly drawn values from a normal distribution with the mean and standard deviation matching the observed PDI scores; (ii) a generalized linear regression model was applied to the new data set to estimate the association between PDI scores and BMI (i.e., mediator model), first using the original variable *A* and then the new variable *A**; (iii) using predicted probabilities from the mediator model with *A* and *A**, we derived the individual stabilized weights as $W_{i}^{c}\;=\;\frac{P\left(M\;=\;M_{i}|A\;=\;A_{i}^{*},\;C\;=\;C_{i}\right)}{P\left(M\;=\;M_{i}|A\;=\;A_{i},\;C\;=\;C_{i}\right)}$, where *C* represents all potential confounders, and *M* the mediator (BMI); and (iv) a weighted logistic regression model was applied to estimate the association between PDI scores (*A* and *A**) and incidence of asthma symptoms (i.e., by modelling the incidence of asthma symptoms according to *A* and *A** in the weighted population). Odds ratios (OR) and 95% confidence intervals (CI) were estimated for an increase of 1 quintile (or tertile) or per 10 points increment in PDI scores; 95% CI were obtained from 500 bootstrap samples. To evaluate the total effect (OR_TE_), we calculated OR_TE_ = OR_DE_ × OR_IE_. When direct and indirect effects operated in the same direction [35], we calculated the “proportion explained” by BMI as (OR_TE_ − OR_DE_)/(OR_TE_ − 1) [36]. 

We assumed that the following conditions were satisfied for the application of mediation analysis [34]: no unmeasured confounder for the associations between (i) PDI scores and incidence of asthma symptoms, (ii) BMI and incidence of asthma symptoms, (iii) PDI scores and BMI; and (iv) no confounders in the BMI-incidence of asthma symptoms association affected by PDI scores. We considered age, smoking, physical activity, education, marital status, and having farmer parents as potential confounders and included those variables in the multivariate models. To account for potential residual confounding by smoking (never vs. ever smokers), we conducted stratified analyses by smoking status and formally tested the statistical significance of the interaction terms between smoking and the PDI scores on asthma incidence.

We also conducted several sensitivity analyses: (1) as in the main analyses, models were not adjusted for total energy intake to avoid an isocaloric substitution among foods [37]; we additionally performed analyses further adjusted for total energy intake; (2) instead of using BMI collected in 2008, we conducted analyses using BMI collected in 2005 or in 2011; (3) we further excluded women with cancers or cardiovascular diseases (CVD) at baseline (1993); and (4) we investigated the association between the “Plant-based Diet Index” (PDI), which does not account for the nutritional quality of the plant-based diet [15], in relation with incident asthma symptoms.

All analyses were performed using SAS version 9.4 (SAS Institute Inc.).

## 3. Results

### 3.1. Participant Characteristics

Women were aged 62 years on average at baseline, and 9.7% of women with no symptoms in 2011 reported at least one asthma symptom in 2018. After adjustment for age, women in the highest quintile of the hPDI consumed less energy, were less likely to be current smokers, had a higher educational level, and were less often overweight or obese, as compared to women in the lowest quintile of the hPDI (Table 1). After adjustment for age, women in the highest quintile of the uPDI consumed less energy, were less physically active, were more likely to be current smokers, and were less often overweight or obese, as compared to women in the lowest quintile of the uPDI (Table 2).

### 3.2. Association between PDI Scores and the Incidence of Asthma Symptoms

#### 3.2.1. *Healthful* Plant-Based Diet Index

Regarding the association between the hPDI and the incidence of asthma symptoms (Table 3), we found a borderline significant inverse total effect (OR (95%CI) per 10 increments of the hPDI = 0.85 (0.73–1.01)) and a significant inverse indirect effect mediated by BMI (OR (95%CI) per 10 increments of the hPDI = 0.94 (0.89–1.00)), with a dose-response relationship (OR (95%CI) = 0.98 (0.95–0.99) for quintile 2 (Q2), 0.96 (0.91–0.98) for quintile 3 (Q3), 0.94 (0.89–0.98) for quintile 4 (Q4), and 0.93 (0.87–0.97) for quintile 5 (Q5) as compared with the first quintile (Q1)). Although the OR was lower than 1, the direct effect of the hPDI on incident asthma symptoms was not statistically significant (OR (95%CI per 10 increments of the hPDI = 0.90 (0.76–1.06)). The proportion of the association between the hPDI and the incidence of asthma symptoms mediated by BMI accounted for one-third (33%) of the total effect. Further adjustment for energy intake (Appendix A), considering BMI in 2005 (Appendix A) or in 2011 (Appendix A), or exclusion of women with cancers or CVD (Appendix A), led to similar findings.

In analyses stratified according to the smoking status, ORs were similar to those of the main results in each category with a significant inverse indirect effect mediated by BMI (ORs (95%CIs) for tertile 3 were 0.91 (0.80–0.98) in never smokers and 0.97 (0.94–0.99) in ever smokers (as compared with tertile 1), and the interaction term between the hPDI and smoking status was not statistically significant *(p* interaction = 0.99) (Figure 2). The proportion mediated by BMI in the association between the hPDI, and the incidence of asthma symptoms represented 40% of the total effect in never smokers and 21% in ever smokers. 

#### 3.2.2. *Unhealthful* Plant-based Diet Index

When the plant-based diet was assessed by the uPDI (Table 4), we reported a significant inverse indirect effect mediated by BMI (OR per 10 increments of the uPDI (95%CI) = 0.92 (0.70–1.00)), with a dose-response relationship (OR (95%CIs) = 0.96 (0.90–0.99) for Q2, 0.93 (0.85–0.98) for Q3, 0.91 (0.83–0.97) for Q4, and 0.90 (0.81–0.96) for Q5, as compared with Q1). The proportion mediated by BMI in the association between the uPDI and incidence of asthma symptoms accounted for 89% of the total effect. The direct and total effects were not statistically significant, but contrary to our hypothesis, ORs were below 1 (OR per 10 increments of the uPDI (95%CI) = 0.91 (0.73–1.09) for total effect and 0.99 (0.81–1.19) for direct effect). Further adjustment for energy intake (Appendix A), considering the BMI in 2005 (Appendix A) or in 2011 (Appendix A), or exclusion of women with cancers or CVD (Appendix A) led to similar results.

In analyses stratified according to the smoking status, ORs for the indirect effect remained below one both in never and ever smokers with a significant inverse indirect effect mediated by BMI (ORs (95%CIs) for tertile 3 were of 0.92 (0.83–0.98) in never smokers and 0.93 (0.88–0.98) in ever smokers (as compared to tertile 1). Although the interaction between uPDI and smoking was not significant *(p* interaction = 0.72), ORs for direct and total effects were above 1 in never smokers only (Figure 3). The proportion mediated by BMI in the association between uPDI and the incidence of asthma symptoms represented 23% of the total effect in ever smokers but could not be calculated in never smokers since direct and indirect effects did not operate in the same direction. 

#### 3.2.3. Plant-Based Diet Index

When the plant-based diet was assessed by the PDI, we also found a significant inverse indirect effect mediated by BMI, with a significant dose-response relationship. Although the ORs were lower than 1, the direct effect of the PDI on incident asthma symptoms was not statistically significant (Appendix A).

## 4. Discussion

In this prospective study of more than 5700 elderly women, better adherence to a plant-based diet was associated with a lower incidence of asthma symptoms over time. One-third of the association was mediated by BMI when women favored healthier plant foods over less healthy plant foods, and almost completely by BMI when women favored less-healthy plant foods over healthier plant foods. Contrary to our hypothesis, we found that adherence to an unhealthy plant-based diet was associated with a lower risk of asthma symptoms. However, the proportion of these associations mediated by BMI was different when we accounted for the nutritional quality of plant-based diets, suggesting that a healthy plant-based diet is more directly related to a lower rate of asthma incidence than an unhealthy plant-based diet, for which the effect is almost totally explained by decreased BMI. Interestingly, and even if not statistically significant, our results among never smokers are consistent with our hypothesis, i.e., that adherence to the uPDI is associated with an increased incidence of asthma symptoms. Overall, our findings suggest that a plant-based diet is associated with a reduced risk of asthma incidence over time, and the risk reduction is partly mediated by BMI. Although the results were not statistically significant, the improved nutritional quality of a plant-based diet may have a direct impact on the incidence of asthma symptoms, especially in never smokers. 

To the best of our knowledge, no study has investigated the relationship between PDI scores and asthma. Using data from the E3N study, we recently reported that both better adherence to the hPDI and the uPDI had direct and indirect effects through BMI on reducing the risk of type 2 diabetes among elderly women [38]. In addition, using generalized structural equation modeling on cross-sectional data, we found that both a healthier diet (AHEI-2010 diet score) and a lower BMI were associated with a lower asthma symptoms score [39]. By contrast, a longitudinal study that investigated the relationship between the AHEI-2010 and changes in asthma symptoms reported that better diet quality was associated with improved asthma symptoms over time in never smokers only, not mediated by BMI [23]. Although dietary scores are different, the discrepancies between the studies could rather be explained by the different populations (middle-aged men and women in the previous study vs. older women), as well as the difference in classification for the reference group. To our knowledge, six other studies investigated the association between different dietary scores (the AHEI-2010 [40,41,42,43], the Mediterranean diet score [43], or the Dietary Inflammatory Index (DII) [44,45]) and asthma outcomes among young [41] to middle-aged adults [40,41,42,43,44,45]. Studies using a dichotomous definition of asthma reported mostly inconclusive associations [40,41,42,44,45], whereas the study based on the continuous asthma symptoms score reported that a healthy diet (evaluated through the AHEI-2010 or the Mediterranean diet score) was associated with lower asthma symptoms [43]. Differences in findings may result from the assessment of asthma (continuous or dichotomous), the assessment of diet (dietary scores based on recommended foods or nutrients for disease prevention or related to pathophysiological processes relevant to asthma), or the role of BMI in the association (confounder vs. mediator). 

Diet, physical activity, and body composition are nutritional factors that are closely interrelated, which makes it difficult to disentangle their separate effects on asthma outcomes. To date, few studies on the role of diet in asthma have adequately addressed the methodological challenges posed by the complexity of these interrelationships. The established evidence that links diet to obesity, and obesity to asthma [17], suggest obesity to be a potential mediator in the diet–asthma association [22] rather than a confounder. The traditional BMI-adjusted approach could over-adjust the association and lead to biased results. Mediation analysis in the counterfactual framework allows considering BMI correctly in such an association; however, it is based on strong assumptions, i.e., that there are no unmeasured confounders for the exposure-outcome, exposure-mediator or mediator-outcome, and there are no variables that are effects of exposure and that confound the relation between mediator-outcome [35]. This approach also provides a quantitative measurement of the proportion mediated through a given mediator, which could help us better understand the potential mechanisms involved in the observed associations. Nevertheless, this measurement is problematic when the natural direct effect and indirect effect operate in different directions since we can either obtain a proportion mediated greater than 100% or the proportion mediated is not calculable (for example, if the total effect is equal to 0) [35]. We acknowledge that mediation analyses in the counterfactual framework usually provide small estimates for indirect effects but allow disentangling the indirect effect from the total effect and interpreting both the strength of the association and the proportion mediated. Although several other methods have now been proposed to compute mediation analyses in the counterfactual framework [33,46], we used the marginal structural model proposed by Lange et al. [34], which can be implemented in standard software for almost any type of variable (e.g., quintiles of dietary scores considered as exposure). 

Several hypotheses and mechanisms have been suggested to explain the role of diet in the development and/or the progression of asthma, including oxidative stress and inflammation [47]. Indeed, asthma is a chronic inflammatory lung disease, and it has been shown that inflammatory agents such as reactive oxygen and/or nitrogen species play a major role in airway inflammation [48] and that asthma is related to a lack of antioxidants increasing susceptibility to oxidative stress inducing a systematic inflammation [48]. A plant-based diet (healthy or unhealthy) has been suggested to reduce pro-inflammatory markers while increasing anti-inflammatory markers [49,50], possibly through the reduction of animal foods such as processed meat; processed meat intake could increase inflammation and oxidative/nitrosative stress in the lungs [51]. Regarding obesity, increasing evidence suggests that elevated levels of inflammatory markers may contribute to the development of obesity [52]. In this manner, both preventing obesity and choosing a plant-based diet could be relevant for the prevention of asthma. 

The role of smoking should be carefully addressed in studies on the diet-asthma association, as it is also likely that this association is impacted by other factors, such as smoking, that may modify the pulmonary oxidant/antioxidant balance [24]. Moreover, the diet-asthma association may result from residual confounding by smoking (e.g., smoking intensity and duration), and smoking is associated with a poorer lifestyle [53]. Although the total and direct effects were not statistically significant, we reported ORs lower than 1 for the association between the hPDI and incidence of symptoms both among never and ever smokers, whereas, for the uPDI, we reported ORs lower than 1 in ever smokers but greater than 1 in never smokers. These results suggest that among ever smokers, which already experiment higher oxidative stress and/inflammation because of tobacco inhalation [54], choosing a plant-based diet regardless of its nutritional quality may be beneficial to reduce the risk of asthma, whereas among never smokers, only choosing healthy plant-based foods may be beneficial, while unhealthy plant-based foods would be deleterious. At least five studies investigated the association between an unhealthy plant-based diet and the risk of chronic diseases according to smoking status [16,55,56,57,58], and only one reported a positive association between the uPDI and the risk of type 2 diabetes in never smokers only [58]. 

Our study has several strengths and limitations. First, our study was longitudinal with a large sample size; this allowed for the accounting of several potential confounders and performing stratified analyses to address the robustness of the findings, although our analytic population represented only one-third of the Asthma-E3N study and subsample sizes were sometimes limited. Secondly, regarding temporality between the exposure, the mediator, and the outcome, we acknowledge some limitations: first, women may have modified their diet between 1993 and 2011; however, we used the mean of the 1993 and 2005 diet scores, estimated from validated food questionnaires [59]; second, asthma symptoms were evaluated in 2011 and 2018, which represents only two measurement points to assess changes in asthma symptoms, but, we used validated tools to estimate the asthma symptoms score [30]; finally, despite the relatively long time period between the assessment of diet and asthma symptoms, BMI was evaluated in-between, in 2008, which respects temporality. BMI may change during the study period; therefore, as we have self-reported weight and, therefore, BMI updated at each questionnaire, we performed sensitivity analyses using BMI in 2005 and 2011 (instead of 2008); findings were very similar to those of the main analyses. Although we acknowledge the limitations of self-reported information on BMI, self-reported anthropometry has proven reliable in a validation study within the E3N cohort [60]. The main source of disease misclassification in this population of elderly women is probably the misdiagnosis of Chronic Obstructive Pulmonary Disease (COPD), and we acknowledge that some potential overlap between asthma and COPD may have contributed to the association between the PDI scores and asthma. However, we used the asthma symptoms score that measures specific symptoms of asthma (and not of COPD) rather than a dichotomous definition of asthma, which is more likely to include COPD patients [30] and may lead to misclassification and biased estimates [61]. In this context, we chose to use the asthma symptom score; it is a multi-categorical measurement based on several asthma symptoms; it is applicable to women both with and without asthma and reflects either the incidence of asthma (among women without asthma) or the temporal variability of the disease among women with asthma. It has been reported to be a more powerful tool for investigating risk factors for asthma [30]. Moreover, regarding diet, recent studies using the asthma symptom score as a continuous definition of asthma have shown evidence of associations, whereas no evidence was found using a binary definition for asthma [22]. As asthma-related multimorbidity is very common, especially among the elderly [20], we further excluded women with previous comorbidities (cancer and cardiovascular diseases) and observed similar associations, suggesting that diet may play a role on asthma beyond its association with other chronic diseases. Regarding the evaluation of PDI scores from food questionnaires, we also acknowledge that limited information was available for whole grains intake, as this type of food was not much eaten between 1990 and the early 2000s; this intake, which is likely underestimated in our study but essential to distinguish hPDI from uPDI, may have contributed to similar results for hPDI and uPDI. Finally, the relative homogeneity of the studied population (i.e., elderly women with mostly high educational levels) actually helps with causal inferences about the relationship between a healthy diet and asthma outcomes because the comparability of the high and low dietary score groups will be higher than in a more heterogeneous population (i.e., less potential for residual confounding). 

## 5. Conclusions

Plant-based diets were associated with a reduced risk of asthma incidence over time, which was partly or almost completely mediated by BMI, depending on their nutritional quality. These results confirm that both preventing obesity and choosing a healthy diet, oriented towards greater consumption of plant-based foods, should be encouraged in multi-intervention programs for the primary prevention of asthma among elderly women.

## Figures and Tables

**Figure 1 nutrients-15-00052-f001:**
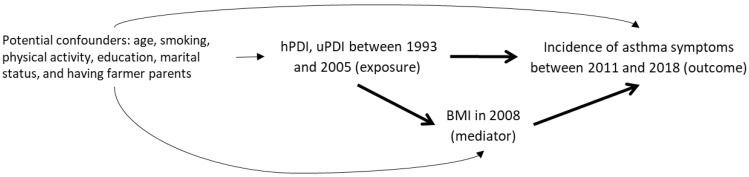
Conceptual model of the association between healthy Plant-based Diet Index (hPDI) and unhealthy Plant-based Diet Index (uPDI) and incidence of asthma symptoms considering BMI as a potential mediator.

**Figure 2 nutrients-15-00052-f002:**
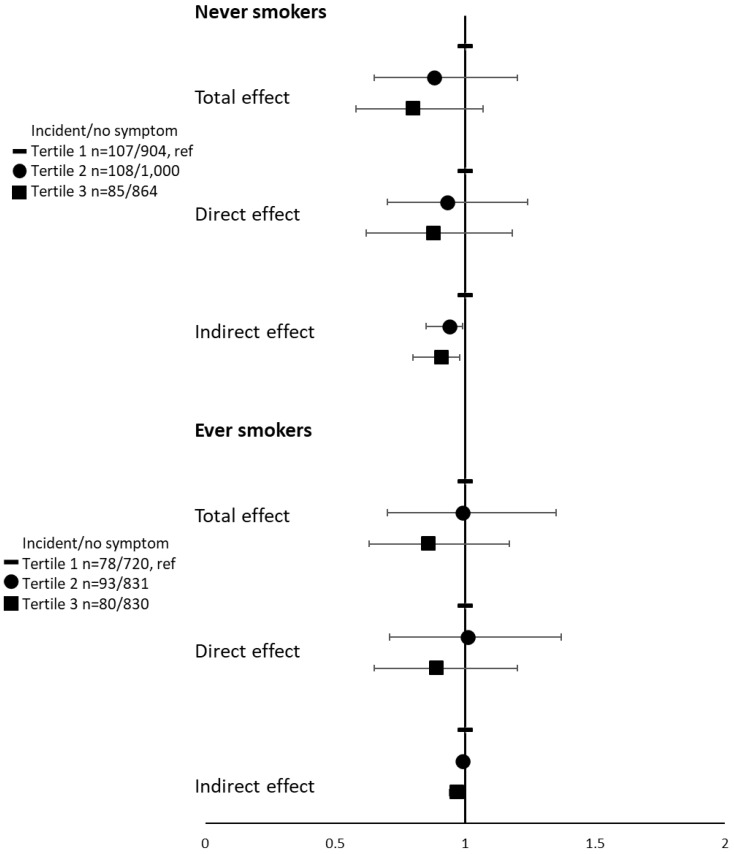
Associations between tertiles of the healthy Plant-based Diet Index (hPDI) and the incidence of asthma symptoms, according to smoking status. Models were adjusted for age, physical activity, educational level, marital status, and having farmer parents. The first tertile (T1) serves as a reference.

**Figure 3 nutrients-15-00052-f003:**
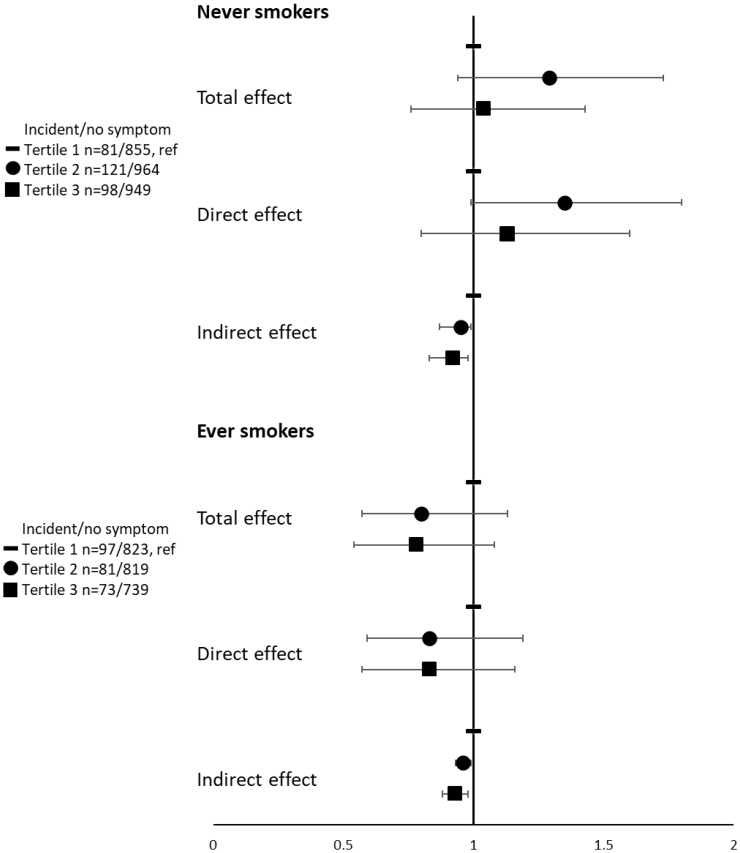
Associations between tertiles of the unhealthy Plant-based diet Index (hPDI) and the incidence of asthma symptoms, according to smoking status. Models were adjusted for age, physical activity, educational level, marital status, and having farmer parents.

**Table 1 nutrients-15-00052-t001:** Baseline characteristics of women according to quintiles of the healthy Plant-based Diet Index (hPDI) (n = 5700).

hPDI Diet Score						
	Quintile 1	Quintile 2	Quintile 3	Quintile 4	Quintile 5	Age-Adjusted *p* for Trend
hPDI diet score, min-max	31–50.5	51–54	54.5–57.5	58–61	61.5–78.5	
hPDI diet score, m(sd)	47.5 (2.7)	52.6 (1.03)	55.9 (1.01)	59.4 (1.00)	64.5 (2.7)	
uPDI diet score, m (sd)	54.0 (5.8)	52.7 (6.0)	51.8 (5.9)	50.7 (5.9)	49.3 (5.6)	
Component score of hPDI (g/d)						
Fiber in whole grains	2.0 (2.5)	2.6 (2.9)	3.3 (3.3)	4.0 (3.2)	4.9 (3.7)	<0.0001
Fruits	223.2 (128.3)	260.3 (142.2)	283.2 (140.2)	312.9 (153.1)	361.1 (161.0)	<0.0001
Vegetables	16.9 (67.7)	29.3 (102.7)	18.3 (78.2)	21.5 (96.1)	28.8 (113.4)	0.01
Nuts	7.0 (8.0)	7.2 (8.7)	6.9 (8.4)	7.7 (9.7)	8.9 (11.9)	<0.0001
Legumes	23.0 (21.4)	23.8 (22.3)	23.7 (22.2)	23.6 (22.5)	27.6 (27.6)	<0.0001
Vegetable oils	21.4 (9.5)	23.2 (10.5)	24.7 (10.3)	25.4 (10.9)	30.2 (12.1)	<0.0001
Tea and coffee	436.7 (243.1)	487.0 (279.9)	509.8 (270.9)	551.2 (294.8)	622.7 (317.3)	<0.0001
Fruit juices	97.2 (92.9)	76.7 (88.2)	67.4 (84.3)	57.3 (74.7)	46.0 (80.6)	<0.0001
Refined grains	223.7 (89.3)	198.0 (94.8)	173.7 (88.4)	151.0 (86.5)	123.4 (73.1)	<0.0001
Potatoes	87.7 (46.7)	72.6 (41.5)	66.3 (42.5)	55.1 (36.8)	46.6 (36.1)	<0.0001
Sugar-sweetened beverages	12.6 (33.3)	7.7 (27.7)	3.7 (15.2)	2.1 (9.3)	1.5 (8.4)	<0.0001
Sweets and desserts	78.7 (40.6)	68.0 (39.1)	62.9 (38.2)	56.1 (37.7)	48.3 (36.8)	<0.0001
Animal fat	22.4 (16.9)	18.9 (16.8)	15.3 (15.2)	13.2 (13.9)	10.2 (12.4)	<0.0001
Dairy	341.2 (164.9)	318.9 (178.6)	305.4 (160.9)	307.0 (176.0)	268.4 (160.8)	<0.0001
Egg	30.0 (18.9)	25.4 (17.5)	22.3 (15.5)	19.5 (15.0)	17.6 (14.3)	<0.0001
Fish or Seafood	44.6 (24.4)	41.9 (25.9)	39.8 (23.9)	37.9 (24.6)	35.7 (25.0)	<0.0001
Meat	127.8 (43.8)	113.1 (42.2)	104.4 (40.1)	94.1 (40.9)	81.6 (42.2)	<0.0001
Miscellaneous animal-based food	104.7 (52.9)	87.1 (48.8)	70.5 (40.8)	62.6 (39.3)	48.5 (34.5)	<0.0001
Age (years)	61.3 (5.3)	61.8 (5.5)	62.3 (5.7)	62.6 (5.5)	62.7 (5.5)	
Energy intake (kcal/d)	2581 (518)	2376 (497)	2233 (463)	2124 (443)	2020 (436)	<0.0001
Leisure-time physical activity (METs/week)	63.9 (48.9)	59.5 (49.7)	61.4 (49.3)	64.6 (53.2)	63.8 (48.7)	0.30
Smoking status						<0.0001
Never smoker	606 (56.2)	628 (56.0)	674 (53.8)	628 (53.8)	532 (49.3)	
Ex-smoker	396 (36.7)	411 (36.6)	499 (39.8)	464 (39.7)	477 (44.2)	
Current smoker	76 (7.1)	83 (7.4)	80 (6.4)	76 (6.5)	70 (6.5)	
Educational level						0.0003
<high school diploma	109 (10.1)	115 (10.3)	116 (9.3)	92 (7.9)	79 (7.3)	
High school to 2-level university	537 (49.8)	549 (48.9)	630 (50.3)	616 (52.7)	507 (47.0)	
3–4-level university	226 (20.1)	234 (20.8)	234 (18.7)	213 (18.2)	225 (20.9)	
≥5-level university	173 (16.1)	188 (16.8)	242 (19.3)	218 (18.7)	234 (21.7)	
Missing	33 (3.1)	36 (3.2)	31 (2.4)	29 (2.5)	34 (3.1)	
Marital status						0.36
No	257 (23.8)	245 (24.0)	302 (24.1)	284 (24.3)	273 (25.3)	
Yes	821 (76.2)	877 (78.2)	950 (75.8)	884 (75.7)	806 (74.7)	
Missing			1 (0.1)			
Having farmer parents						0.42
No	941 (87.3)	959 (85.4)	1086 (86.7)	1001 (85.7)	921 (85.4)	
Yes	109 (10.1)	143 (12.8)	142 (11.3)	149 (12.8)	125 (11.6)	
Missing	28 (2.6)	20 (1.8)	25 (2.0)	18 (1.5)	33 (3.0)	
BMI (kg/m^2^)						<0.0001
<20	147 (13.6)	185 (16.5)	202 (16.1)	223 (19.1)	233 (21.6)	
20–24.9	614 (57.0)	656 (58.5)	732 (58.4)	685 (58.6)	617 (57.2)	
25–29.9	261 (24.2)	240 (21.4)	265 (21.2)	217 (18.6)	194 (18.0)	
≥30	56 (5.2)	41 (3.6)	54 (4.3)	43 (3.7)	35 (3.2)	
BMI (kg/m^2^)	23.6 (3.9)	23.2 (3.6)	23.3 (4.0)	23.0 (3.5)	22.7 (3.6)	<0.0001

Data are presented as n (%) or mean (SD) unless otherwise stated. *p* for trend was calculated using the quintile median values.

**Table 2 nutrients-15-00052-t002:** Baseline characteristics of women according to quintiles of the unhealthy Plant-based Diet Index (uPDI) (n = 5700).

uPDI Diet Score						
	Quintile 1	Quintile 2	Quintile 3	Quintile 4	Quintile 5	Age-Adjusted *p* for Trend
uPDI diet score, min-max	31–46.5	47–50.5	51–53.5	54–57	57.5–72.5	
uPDI diet score, m(sd)	43.4 (2.7)	48.8 (1.16)	52.2 (0.86)	55.5 (1.02)	60.5 (2.7)	
hPDI diet score, m(sd)	58.0 (2.7)	56.6 (6.0)	55.9 (6.0)	54.7 (5.7)	53.4 (5.5)	
Component score of uPDI (g/d)						
Fiber in whole grains	4.7 (3.6)	4.0 (3.4)	3.1 (3.1)	2.8 (3.1)	1.9 (2.4)	<0.0001
Fruits	349.0 (153.9)	317.7 (150.5)	291.4 (146.7)	258.3 (143.8)	219.0 (130.9)	<0.0001
Vegetables	36.9 (127.8)	19.4 (85.3)	18.4 (83.4)	17.0 (72.2)	23.2 (86.1)	0.0001
Nuts	10.5 (10.7)	8.3 (9.6)	7.4 (9.7)	6.1 (7.7)	4.3 (6.4)	<0.0001
Legumes	34.1 (27.0)	27.6 (24.3)	24.0 (23.2)	19.0 (18.2)	16.1 (17.9)	<0.0001
Vegetable oils	33.0 (10.9)	27.6 (10.7)	24.7 (9.8)	21.5 (8.8)	17.4 (8.0)	<0.0001
Tea and coffee	656.7 (317.2)	538.9 (279.5)	516.4 (287.0)	466.6 (255.8)	424.7 (245.3)	<0.0001
Fruit juices	60.1 (85.8)	68.4 (82.4)	66.1 (79.1)	70.8 (88.8)	78.4 (92.4)	<0.0001
Refined grains	157.1 (90.6)	164.8 (92.8)	176.2 (96.9)	179.3 (93.0)	193.4 (89.0)	<0.0001
Potatoes	65.0 (44.1)	65.3 (45.3)	63.8 (40.1)	64.9 (41.4)	68.8 (44.1)	0.07
Sugar-sweetened beverages	2.7 (13.7)	3.9 (17.5)	4.7 (16.9)	5.1 (16.4)	11.1 (35.3)	<0.0001
Sweets and desserts	57.1 (39.4)	60.6 (39.6)	62.3 (40.0)	63.7 (37.2)	70.4 (41.5)	<0.0001
Animal fat	23.1 (19.9)	17.4 (15.3)	15.1 (14.5)	13.1 (13.1)	10.7 (11.2)	<0.0001
Dairy	379.1 (185.8)	331.2 (172.0)	305.4 (165.3)	279.2 (146.4)	241.3 (141.7)	<0.0001
Egg	32.5 (18.9)	25.2 (17.5)	22.1 (15.4)	19.2 (13.9)	15.2 (12.2)	<0.0001
Fish or Seafood	55.8 (28.3)	45.1 (25.9)	37.2 (21.0)	33.7 (19.7)	26.7 (16.9)	<0.0001
Meat	122.3 (47.4)	109.2 (45.5)	103.4 (42.1)	95.2 (40.6)	89.4 (38.9)	<0.0001
Miscellaneous animal-based food	89.0 (54.1)	76.6 (47.0)	73.0 (45.6)	69.3 (45.4)	63.8 (41.7)	<0.0001
Age (years)	61.5 (5.1)	62.2 (5.5)	62.2 (5.5)	62.3 (5.6)	62.7 (6.0)	
Energy intake (kcal/d)	2553 (528)	2355 (495)	2238 (471)	2129 (443)	2027 (436)	<0.0001
Leisure-time physical activity (METs/week)	67.7 (54.1)	64.3 (50.4)	61.0 (46.2)	61.7 (48.1)	58.2 (50.6)	<0.0001
Smoking status						<0.0001
Never smoker	527 (47.7)	710 (54.3)	591 (55.0)	613 (54.7)	627 (57.4)	
Ex-smoker	502 (45.5)	513 (39.3)	421 (39.2)	439 (39.2)	372 (34.0)	
Current smoker	75 (6.8)	84 (6.4)	63 (6.9)	69 (6.2)	94 (8.6)	
Educational level						0.33
<high school diploma	100 (9.1)	111 (8.6)	89 (8.3)	95 (8.5)	115 (10.5)	
High school to 2-level university	546 (49.4)	657 (50.3)	547 (50.9)	571 (51.0)	518 (47.4)	
3–4-level university	216 (19.6)	259 (19.8)	217 (20.2)	212 (18.9)	228 (20.9)	
≥5-level university	206 (18.6)	232 (17.7)	197 (18.3)	219 (19.5)	201 (18.4)	
Missing	36 (3.3)	47 (3.6)	25 (2.3)	24 (2.1)	31 (2.8)	
Marital status						0.84
No	263 (23.8)	298 (22.8)	272 (25.3)	259 (23.1)	269 (24.6)	
Yes	841 (76.2)	1009 (77.2)	803 (74.7)	861 (76.8)	824 (75.4)	
Missing				1 (0.09)		
Having farmer parents						0.73
No	931 (84.3)	1120 (85.7)	929 (86.4)	967 (86.3)	961 (87.9)	
Yes	150 (13.6)	153 (11.7)	126 (11.7)	133 (11.8)	106 (9.7)	
Missing	23 (2.1)	34 (2.6)	20 (1.9)	21 (1.9)	26 (2.4)	
BMI (kg/m^2^)						<0.0001
<20	148 (13.4)	195 (15.0)	201 (18.7)	218 (19.5)	228 (20.8)	
20–24.9	623 (56.4)	751 (57.5)	611 (56.8)	669 (59.7)	650 (59.5)	
25–29.9	265 (24.0)	304 (22.3)	218 (20.3)	200 (17.8)	190 (17.4)	
≥30	68 (6.2)	57 (4.4)	45 (4.2)	34 (3.0)	25 (2.3)	
BMI (kg/m^2^)	23.8 (3.9)	23.4 (3.8)	23.2 (3.9)	22.7 (3.5)	22.6 (3.5)	<0.0001

Data are presented as n (%) or mean (SD) unless otherwise stated. *p* for trend was calculated using the quintile median values.

**Table 3 nutrients-15-00052-t003:** Association between the healthy Plant-based Diet Index (hPDI) and the incidence of asthma symptoms, mediated by BMI.

hPDI	No.	Total Effect	Direct Effect	Indirect Effect	Proportion Mediated
		OR (95%CI)	OR (95%CI)	OR (95%CI)	
Continuous	551/5149	0.85 (0.73–1.01)	0.90 (0.76–1.06)	0.94 (0.89–1.00)	33%
Quintile 1	107/969	1.00 (ref)	1.00 (ref)	1.00 (ref)	40%
Quintile 2	108/1008	0.85 (0.63–1.11)	0.87 (0.65–1.14)	0.98 (0.95–0.99)	
Quintile 3	127/1119	0.87 (0.66–1.14)	0.91 (0.69–1.20)	0.96 (0.91–0.98)	
Quintile 4	110/1055	0.84 (0.62–1.09)	0.89 (0.66–1.16)	0.94 (0.89–0.98)	
Quintile 5	99/998	0.85 (0.62–1.11)	0.91 (0.66–1.22)	0.93 (0.87–0.97)	

Ref = referent values. Odds ratio (OR) and 95% confidence interval (CI) were estimated from marginal structural models for an increase of one quintile in the hPDI or per 10 increments of hPDI; 95% CI was obtained from 500 bootstrapped samples. The total effect represents the overall effect of the exposure (diet) on the disease (asthma); the indirect effect represents the effect passing through the mediator (body mass index); the direct effect represents the effect unexplained by the mediator. Models were adjusted for age, physical activity, smoking, educational level, marital status, and having farmer parents.

**Table 4 nutrients-15-00052-t004:** Association between the unhealthy Plant-based Diet Index (uPDI) and the incidence of asthma symptoms, mediated by BMI.

uPDI	No.	Total Effect	Direct Effect	Indirect Effect	Proportion Mediated
		OR (95%CI)	OR (95%CI)	OR (95%CI)	
Continuous	551/5149	0.91 (0.73–1.09)	0.99 (0.81–1.19)	0.92 (0.70–1.00)	89%
Quintile 1	106/1017	1.00 (ref)	1.00 (ref)	1.00 (ref)	83%
Quintile 2	134/1138	1.07 (0.82–1.39)	1.11 (0.86–1.46)	0.96 (0.90–0.99)	
Quintile 3	98/973	0.91 (0.65–1.23)	0.97 (0.72–1.31)	0.93 (0.85–0.98)	
Quintile 4	109/1016	0.99 (0.73–1.30)	1.09 (0.77–1.45)	0.91 (0.83–0.97)	
Quintile 5	104/1005	0.88 (0.65–1.19)	0.98 (0.73–1.32)	0.90 (0.81–0.96)	

Ref = referent values. Odds ratio (OR) and 95% confidence interval (CI) were estimated from marginal structural models for an increase of 1 quintile in the uPDI or per ten increments of uPDI; 95% CI was obtained from 500 bootstrapped samples. The total effect represents the overall effect of the exposure (diet) on the disease (asthma); the indirect effect represents the effect passing through the mediator (body mass index); the direct effect represents the effect unexplained by the mediator. Models were adjusted for age, physical activity, smoking, educational level, marital status, and having farmer parents.

## Data Availability

Requests for access to data, statistical code, questionnaires, and technical processes may be made by contacting the corresponding author at nasser.laouali@inserm.fr.

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
