# Peer review of "Plant-Based Diets and the Incidence of Asthma Symptoms among Elderly Women, and the Mediating Role of Body Mass Index"

_nutrients, 2022, doi:10.3390/nu15010052_

Round 1
Reviewer 1 Report
Dear Editor,
I was asked to review the manuscript with the title “Plant-Based Diets and the Incidence of Asthma Symptoms 2 among Elderly Women, and the Mediating Role of Body Mass 3 Index”. The coauthors wanted to investigate the influence of healthy and unhealthy plant-based diets on the asthma outcome and whether there would be a potential association. As a result, both plant-based diets were associated with a reduced risk of asthma incidence over time in 5,700 asthmatic women. This association was partly or completely mediated by BMI.
The manuscript is interesting to read. I have no further comments.
Author Response
We thank the reviewer for the careful reading. In order to clarify our research design, as also suggested by R2, we modified the method section accordingly (please see R2 and R3 in the response to reviewer 2).

Reviewer 2 Report
The study by Ait-hadad et al. aims to investigate the association between plant-based diet (healthy and unhealthy) and asthma symptoms and to evaluate the potential mediating role of BMI. The manuscript is well-written, and the study has several strengths including the repeated measurements of exposure and outcome and the authors performed several sensitivity analyses.
My main comment relates to the long-time gap between the diet assessment in 1993 and 2005 and the incidence of asthma symptoms starting from 2011 to 2018. If I understood correctly, individuals with asthma symptoms in 2011 were excluded, which may not be optimal since low PDI in 1993-2005 may already have caused asthma symptoms in 2011. Therefore, you may consider to also analyze prevalent asthma in 2011 and 2018 as sensitivity analyses.
Another limitation is the self-reported information on BMI, which generally tend to be under-reported, especially in overweight/obese individuals. This should be mentioned in the discussion. Also, BMI may change during the study period and the lack of updated information on BMI could also be acknowledged.
Do you have any information on doctors diagnosed asthma or asthma medication use? Self-reported asthma symptoms are subjective and may lead to misclassification. This potential bias including its potential direction could be discussed more.
